# The global burden of *Klebsiella pneumoniae*-associated lower respiratory infection in 204 countries and territories, 1990–2021: Findings from the global burden of disease study 2021

**Juanjuan Li, Liang Xu, A. Fang Zuo, Ping Xu, Kaizhi Xu** ✆*

Emergency & Intensive Care Unit Center, Department of Intensive Care Unit, Zhejiang Provincial People's Hospital, Affiliated People's Hospital, Hangzhou Medical College,Hangzhou, People's Republic of China

* kaizhixu0109@163.com

## Abstract

This study investigates the global epidemiological burden of lower respiratory infections (LRI) attributable to *Klebsiella pneumoniae* from 1990–2021, using data from the Global Burden of Disease Study (GBD) 2021. The findings reveal that globally, disability-adjusted life years (DALYs) from *Klebsiella pneumoniae*-associated LRI decreased from 16,701,044 (95% UI: 14,220,055–19,183,469) in 1990–6,935,440 (95% UI: 5,953,328–8,007,786) in 2021, while deaths declined from 239,367 (95% UI: 212,553–268,072) –175,783 (95% UI: 158,749–193,924). The age-standardized DALYs rate dropped from 313.1 (95% UI: 266.6–359.7)–87.9 (95% UI: 75.4–101.5), and the death rate decreased from 4.5 (95% UI: 4.0–5.0)–2.2 (95% UI: 2.0–2.5). In 2021, the highest rates were observed in Oceania and Sub-Saharan Africa, particularly in Central African Republic, Niger, and Zimbabwe, while the lowest rates were found in Australasia, High-income North America, Eastern Europe, and East Asia, especially in the UAE, Australia, and Qatar. Higher rates were noted among both males and females under 10 and over 65 years old. Although most regions experienced decreases in age-standardized rates(ASR) from 1990–2021, Southern Latin America exhibited an increase. Additionally, age-standardized DALYs and death rates generally declined with increasing socio-demographic index (SDI). The global burden of LRI due to *Klebsiella pneumoniae* significantly decreased over the study period, but lower SDI regions, children, and the elderly remain vulnerable and require targeted interventions to further reduce this burden.

## Introduction

Lower respiratory infections (LRI) continue to be a leading cause of morbidity and mortality globally, despite advances in medical science [1]. LRI encompass a range of infections, including pneumonia, bronchitis, and bronchiolitis, and are primarily

**Data availability statement:** All relevant data for this study are publicly available from the figshare repository (https://doi.org/10.6084/m9.figshare.28616111.v2).

**Funding:** The author(s) received no specific funding for this work.

**Competing interests:** The authors have declared that no competing interests exist.

caused by bacteria, viruses, and fungi [2]. The Global Burden of Disease (GBD) Study 2021 provides a comprehensive analysis of the incidence and mortality burden of LRI from 1990–2021. This study aims to evaluate the global, regional, and national burden of *Klebsiella pneumoniae*-associated LRI from 1990–2021,.*Klebsiella pneumoniae* is a significant pathogen in the context of LRI,Which is a Gram-negative, encapsulated bacterium that has emerged as a formidable pathogen in both community and healthcare settings. It is notorious for causing severe infections, including pneumonia, bloodstream infections, wound infections, and urinary tract infections [3]. The bacterium's ability to acquire and disseminate antibiotic resistance genes has made it a significant public health threat, particularly in hospital environments where it can form biofilms on medical devices, rendering standard treatments less effective [4].

*Klebsiella pneumoniae* is one of the leading bacterial causes of LRI globally. In 2021, *Klebsiella pneumoniae* was responsible for an estimated 176,000 deaths due to LRI, highlighting its critical role in the global LRI burden. This places *Klebsiella pneumoniae* third among the bacterial causes of LRI deaths, following Streptococcus pneumoniae and Staphylococcus aureus, which accounted for 505,000 and 424,000 deaths respectively [5].The high burden of *Klebsiella pneumoniae*-associated LRI can be attributed to several factors: *Klebsiella pneumoniae*'s ability to form biofilms on medical devices such as catheters and ventilators protects the bacteria from antibiotics and the host immune response, leading to persistent infections in hospitalized patients [6]. The widespread emergence of antibiotic-resistant *Klebsiella pneumoniae* strains, including those producing extended-spectrum beta-lactamases (ESBLs) and carbapenemases, complicates treatment and increases the risk of infection spread within healthcare facilities [7]. *Klebsiella pneumoniae* is easily transmitted in healthcare settings through contaminated hands of healthcare workers, medical equipment, and surfaces. Poor infection control practices and overcrowded healthcare facilities exacerbate the spread of *Klebsiella pneumoniae*. Hospitalized patients, particularly those in intensive care units (ICUs), are often immunocompromised or have underlying health conditions, making them more susceptible to infections [8]. The use of invasive procedures and prolonged hospital stays further increases the risk of acquiring *Klebsiella pneumoniae* infections [9].

*Klebsiella pneumoniae* exhibits significant antibiotic resistance, which complicates treatment options. It produces ESBLs and carbapenemases, making it resistant to a wide range of antibiotics, including those typically used as last-resort treatments [10]. The World Health Organization (WHO) has identified *Klebsiella pneumoniae* as a critical priority pathogen due to its antibiotic resistance [11]. The resistance mechanisms in *Klebsiella pneumoniae* include the production of enzymes that degrade antibiotics, alterations in antibiotic target sites, and the use of efflux pumps to expel antibiotics from bacterial cells [12].

The global distribution of *Klebsiella pneumoniae* is influenced by several factors, including healthcare practices, patient demographics, and geographic variations [13]. *Klebsiella pneumoniae* is particularly prevalent in low- and middle-income countries (LMICs), where healthcare infrastructure may be less robust, leading to higher

rates of hospital-acquired infections (HAIs) and community-acquired infections [5].In high-income countries, the burden of *Klebsiella pneumoniae* infections is significant and driven by the widespread use of invasive medical devices, the high prevalence of comorbid conditions in the patient population, and the misuse and overuse of antibiotics [5]. The overuse of antibiotics promotes the development of resistant strains of *Klebsiella pneumoniae*, making infections harder to treat and control.

Given the significant global burden of *Klebsiella pneumoniae*-associated LRI and the challenges posed by antibiotic resistance, this study aims to provide a comprehensive analysis of the epidemiology of *Klebsiella pneumoniae*-associated LRI from 1990–2021. Utilizing data from the Global Burden of Disease (GBD) Study 2021, we evaluate trends in incidence, mortality, and disability-adjusted life years (DALYs) across 204 countries and territories. Existing studies are often limited, focusing on specific regions or short timeframes, and lacking a global perspective. Additionally, many do not comprehensively address the demographic and geographic factors influencing *Klebsiella pneumoniae*-associated LRI. This lack of comprehensive data hinders our understanding of the global impact and trends of *Klebsiella pneumoniae* infections, complicating the development of effective public health strategies. Therefore, this study seeks to fill these gaps by providing detailed, long-term, and globally representative data, offering insights into potential strategies for mitigating the impact of this pathogen on global public health.

## Methods

### Data sources

The data for disability-adjusted life years (DALYs), death counts, age-standardized DALY rates, and age-standardized death rates attributable to *Klebsiella pneumoniae* lower respiratory infection (LRI) from 1990–2021 were sourced from the Global Health Data Exchange (GHDx) query tool (http://ghdx.healthdata.org/gbd-results-tool).

### Definitions

To measure the burden of LRI attributable to *Klebsiella pneumoniae*, we used DALYs and death counts. DALYs are the sum of years lived with disability and years of life lost due to premature death, thereby encapsulating both non-fatal and fatal health impacts. Years of life lost are computed based on standard life expectancy, while years lived with disability are calculated by multiplying the number of individuals with a given condition by the disability weight, which reflects the severity of health loss. Disability weights range from 0 (indicating full health) to 1 (indicating death) [14].

The Sociodemographic Index (SDI) evaluates a country's level of sociodemographic development by integrating three components: income per capita, educational attainment, and total fertility rate (TFR). The SDI scale ranges from 0–1, with 0 representing the lowest levels of income and education, coupled with the highest fertility rates, while 1 signifies the highest levels of income and education, along with the lowest fertility rates. Based on their SDI scores for 2021, countries and regions were classified into five categories: low (0 < 0.46), low-middle (0.46 ≤ 0.61), middle (0.61 ≤ 0.69), high-middle (0.69 ≤ 0.81), and high (0.81 ≤ 1.00). The data cover 204 countries and territories, further grouped into 21 geographical regions.

Since this study is based on de-identified and publicly available data from the Global Burden of Disease (GBD) Study 2021, no direct recruitment of participants or collection of identifiable data occurred. As such, informed consent was not applicable, and no separate ethical approval for participant consent was required. All data used adhered to the ethical standards and data usage policies of the GBD database.

### Statistical analyses

We quantified the LRI burden attributable to *Klebsiella pneumoniae* using ASR, estimated annual percentage change (EAPC), case numbers, and changes in case numbers for DALYs and deaths, disaggregated by global, age, sex, geographical location, and socio-economic status [15]. Changes in case numbers are derived by subtracting the 1990 values

from the 2021 values and dividing by the 1990 values. All estimates are presented with 95% uncertainty intervals (UIs), calculated from 1000 draw-level estimates for each parameter. The 95% UI is the interval between the 25th and 975th values of these draws.

EAPC summarizes trends in ASR over time. An EAPC with a 95% CI entirely above zero indicates an increasing trend, while an EAPC with a 95% CI entirely below zero indicates a decreasing trend. Otherwise, the trend is considered stable. We also examined the relationship between LRI burden attributable to *Klebsiella pneumoniae* and SDI by analyzing 2021 data and trends from 1990–2021, employing local estimated scatterplot smoothing regression across 21 GBD regions and 204 countries and territories. Statistical analyses were performed using R software (Version 4.2.2, R core team).

## Result

Globally, the disability-adjusted life years (DALYs) due to *Klebsiella pneumoniae* associated lower respiratory infection (LRI) significantly decreased from an estimated 16,701,043.599 (95% UI: 14,220,055.375–19,183,468.938) in 1990–6,935,439.843 (95% UI: 5,953,327.570–8,007,785.632) in 2021. Correspondingly, the number of deaths declined from 239,366.782 (95% UI: 212,552.572–268,071.566) in 1990–175,783.213 (95% UI: 158,749.052–193,923.835) in 2021. The age-standardized DALYs rate dropped from 313.128 (95% UI: 266.649 to 359.670) to 87.887(95% UI: 75.441 to 101.475), with an estimated annual percentage change (EAPC) of −3.941 (95% UI: −4.009 to −3.872), while the age-standardized death rate decreased from 4.488 (95% UI: 3.985 to 5.026) to 2.228(95% UI: 2.012 to 2.457), with an EAPC of −2.244 (95% UI: −2.369 to −2.119) (Tables 1, 2). Regarding sex-specific trends, males experienced a reduction in DALYs from 8,726,665.120 (95% UI: 7,385,521.295 to 10,130,890.439) in 1990–3,813,370.463 (95% UI: 3,255,438.803 to 4,459,572.607) in 2021, while deaths decreased from 124,922 (95% UI: 109,896–140,984) to 94,252 (95% UI: 85,746–104,057). The age-standardized DALYs rate for males declined from 324.9 (95% UI: 275.0 to 377.2) to 96.3 (95% UI: 82.2 to 112.6).The age-standardized death rate dropped from 4.65 (95% UI: 4.09 to 5.25) to 2.38 (95% UI: 2.17 to 2.63). Similarly, females showed a decline in both DALYs and deaths, following similar trends to those observed in males(Tables 1–2 and Fig 1). Notably, higher ASR and numbers of DALYs and deaths were observed in both males and females under 10 years old and over 65 years old (Fig 2).

Geographically, in 2021, the regions with the highest age-standardized DALYs and death rates of LRI attributable to *Klebsiella pneumoniae* were Oceania and Sub-Saharan Africa(Tables 1, 2), particularly in countries such as Central African Republic, Niger and Zimbabwe (S1 and S2 Tables). Conversely, the regions with the lowest age-standardized DALYs and death rates of LRI attributable to *Klebsiella pneumoniae* were Australasia,High-income North America,Eastern Europe and East Asia (Tables 1, 2), notably in countries including United Arab Emirates, Australia, and Qatar (S1 and S2 Tables). Additionally, Western Sub-Saharan Africa and South Asia, especially in countries such as India and Nigeria, had the highest absolute numbers of DALYs and death cases of LRI attributable to *Klebsiella pneumoniae* (Figs 3a and 4a).

From 1990 to 2021, most regions experienced a decrease in age-standardized DALYs and death rates, with an EAPC less than 0. However, Southern Latin America showed a significant increase in both age-standardized DALYs and death rates, with an EAPC greater than 0. Additionally, high-income Asia Pacific and Central Europe also exhibited an increasing trend in death rates (Tables 1, 2). Countries such as United States of America, China, Mongolia saw the largest decreases in the EAPC of LRI attributable to *Klebsiella pneumoniae*, while countries including Poland, Italy, Argentina experienced the largest increases (Figs 3b and 4b).Over the same period, The most pronounced increases in DALYs and death cases of LRI attributable to Klebsiella pneumoniae were observed in Southeast Asia, Southern Latin America, and Central Europe.Specifically, countries such as Thailand, Argentina, and Poland experienced the largest increases in DALYs and death cases, while countries like China, Mongolia, and Kazakhstan showed the greatest decreases in DALYs and death cases of LRI attributable to *Klebsiella pneumoniae* (Figs 3c and 4c).

Figs 5a and 6a illustrate the relationship between age-standardized DALYs and death rates and SDI across all regions from 1990 to 2021. In 2021, the highest age-standardized DALYs and death rates of LRI attributable to *Klebsiella*

**Table 1. The number and ASR of DALYs for *Klebsiella pneumoniae*-related LRI burden in 1990 and 2021, and its trends from 1990 to 2021.**

| | 1990 | | 2021 | | 1990–2021 |
| --- | --- | --- | --- | --- | --- |
| | Count,1990 | Age-standardised rate per 100000 population,1990 | Count,2021 | Age-standardised rate per 100000 population,2021 | Estimated annual percentage change,1990–2021 |
| **Sex** | | | | | |
| Male | 8726665.120(7385521.295–10130890.439) | 324.926(274.990–377.210) | 3813370.463(3255438.803–4459572.607) | 96.312(82.221–112.633) | −3.810(−3.876–3.744) |
| Female | 7974378.478(6769262.751–9223036.261) | 301.161(255.648–348.318) | 3122069.380(2659736.014–3586216.191) | 79.402(67.644–91.207) | −4.088(−4.164–−4.011) |
| **Region** | | | | | |
| Global | 16701043.599(14222055.375–19183468.938) | 313.128(266.649–359.670) | 6935439.843(5953327.570–8007785.632) | 87.887(75.441–101.475) | −3.941(−4.009–3.872) |
| Australasia | 2919.655(2702.763–3088.240) | 14.399(13.330–15.231) | 3341.777(2837.297–3675.944) | 10.793(9.164–11.873) | −0.775(−1.206–0.342) |
| Eastern Europe | 82524.950(78326.267–86975.615) | 36.436(34.583–38.401) | 61789.428(56018.698–67758.799) | 29.885(27.094–32.772) | −0.790(−1.617–0.045) |
| Oceania | 37669.366(30138.191–46947.063) | 575.109(460.129–716.755) | 50978.423(39374.979–65120.356) | 366.025(282.712–467.564) | −1.328(−1.453–−1.203) |
| Central Europe | 75264.438(71584.930–79072.024) | 60.167(57.226–63.211) | 48047.269(44414.730–51485.814) | 41.685(38.533–44.668) | −1.123(−1.687–0.555) |
| High-income North America | 79888.015(73741.355–83913.934) | 28.389(26.204–29.819) | 65519.490(58868.928–69847.681) | 17.727(15.903–18.869) | −1.826(−2.080–1.571) |
| High-income Asia Pacific | 70885.633(65543.081–74529.068) | 40.885(37.803–42.986) | 76410.011(64274.696–83842.278) | 41.203(34.659–45.211) | 0.204(−0.044–0.452) |
| Southeast Asia | 1409561.935(1201486.047–1665994.154) | 302.801(258.103–357.888) | 529929.428(455971.553–602308.558) | 75.888(65.297–86.253) | −4.481(−4.632–4.330) |
| Western Europe | 105638.153(97790.893–110797.185) | 27.480(25.439–28.822) | 100387.088(85921.315–108487.130) | 22.951(19.644–24.803) | −0.992(−1.325–0.658) |
| Central Asia | 333707.804(302929.940–367290.413) | 481.459(437.054–529.910) | 106283.801(89362.037–126167.123) | 110.933(93.271–131.686) | −5.007(−5.239–4.776) |
| Southern Latin America | 31048.498(29449.114–32596.553) | 62.676(59.447–65.801) | 47979.843(43609.402–51402.453) | 70.877(64.421–75.933) | 1.315(1.069–1.560) |
| Tropical Latin America | 226310.391(203179.863–252921.422) | 148.350(133.188–165.794) | 133132.811(121355.206–142275.072) | 58.514(53.337–62.532) | −2.550(−3.127–1.970) |
| East Asia | 2471417.722(2124528.046–2865684.855) | 203.001(174.508–235.386) | 302474.185(257561.220–350948.960) | 20.538(17.488–23.829) | −8.262(−8.938–7.580) |
| Andean Latin America | 167346.405(143923.510–191635.095) | 440.462(378.812–504.390) | 55061.482(45075.331–67089.545) | 83.258(68.158–101.446) | −4.939(−5.309–4.568) |
| Southern Sub-Saharan Africa | 186054.491(162662.600–210419.123) | 354.943(310.317–401.424) | 167032.811(141559.684–192934.836) | 208.001(176.280–240.256) | −1.175(−1.625–0.724) |
| Western Sub-Saharan Africa | 2211261.125(1742269.157–2668568.419) | 1144.789(901.988–1381.541) | 1739952.595(1281534.108–2233339.159) | 355.217(261.629–455.944) | −5.265(−5.408–5.121) |
| Caribbean | 80797.551(67247.756–97018.176) | 228.937(190.544–274.897) | 48549.757(38990.714–58728.384) | 102.299(82.157–123.746) | −2.394(−2.666–2.120) |
| Central Latin America | 286748.997(261916.654–312652.403) | 174.413(159.309–190.169) | 122676.200(106310.836–143359.965) | 48.488(42.020–56.664) | −3.995(−4.503–3.485) |
| South Asia | 5326070.880(4382061.240–6212615.384) | 487.105(400.769–568.186) | 1878884.193(1591922.150–2173624.233) | 101.750(86.210–117.711) | −4.758(−4.886–4.630) |
| North Africa and Middle East | 1100105.636(933794.764–1373007.368) | 324.331(275.300–404.788) | 298807.908(254699.780–346627.978) | 47.963(40.883–55.639) | −5.661(−5.738–5.584) |
| Eastern Sub-Saharan Africa | 1892258.203(1518767.007–2355794.621) | 991.620(795.896–1234.532) | 836127.043(671619.033–1008760.290) | 196.229(157.621–236.744) | −5.265(−5.408–5.121) |
| Central Sub-Saharan Africa | 523563.751(391886.573–652512.141) | 952.580(713.005–1187.191) | 261974.302(205359.039–322732.814) | 191.322(149.976–235.695) | −5.150(−5.454–4.845) |

**GBD Global Burden of Disease Study, DALYs disability-adjusted life years, ASR Age-Standardized Rate**

**Table 2. The number and ASR of deaths for *Klebsiella pneumoniae*-related LRI burden in 1990 and 2021, and its trends from 1990 to 2021.**

| | 1990 | | 2021 | | 1990-2021 |
|---|---|---|---|---|---|
| | Number of cases,1990 | Age-standardised rate per 100000 population,1990 | Number of cases,2021 | Age-standardised rate per 100000 population,2021 | Estimated annual percentage change,1990–2021 |
| **Sex** | | | | | |
| Male | 124921.758(109895.861-140983.913) | 4.651(4.092-5.249) | 94252.236(85746.460-104057.424) | 2.380(2.166-2.628) | -2.167(-2.288-2.046) |
| Female | 114445.024(99960.033-129045.252) | 4.322(3.775-4.874) | 81530.977(70278.970-91724.124) | 2.074(1.787-2.333) | -2.328(-2.458-2.198) |
| **Region** | | | | | |
| Global | 239366.782(212552.572-268071.566) | 4.488(3.985-5.026) | 175783.213(158749.052-193923.835) | 2.228(2.012-2.457) | -2.244(-2.369-2.119) |
| Australasia | 163.314(145.335-176.292) | 0.805(0.717-0.869) | 270.607(221.382-302.100) | 0.874(0.715-0.976) | 0.301(-0.130-0.733) |
| Central Europe | 1929.250(1842.275-2006.031) | 1.542(1.473-1.604) | 2485.489(2254.273-2669.764) | 2.156(1.956-2.316) | 1.067(0.564-1.572) |
| Eastern Europe | 1523.282(1460.286-1584.878) | 0.673(0.645-0.700) | 1983.545(1796.894-2169.968) | 0.959(0.869-1.050) | 0.826(-0.067-1.728) |
| Oceania | 476.658(387.026-580.312) | 7.277(5.909-8.860) | 686.776(548.965-858.150) | 4.931(3.942-6.162) | -1.152(-1.252-1.052) |
| High-income Asia Pacific | 4122.699(3693.390-4362.700) | 2.378(2.130-2.516) | 6296.811(5061.529-7005.948) | 3.395(2.729-3.778) | 1.287(1.040-1.535) |
| Southeast Asia | 19464.946(16710.022-22385.634) | 4.181(3.590-4.809) | 16866.363(14368.396-18776.887) | 2.415(2.058-2.689) | -1.678(-1.932-1.422) |
| High-income North America | 4599.750(4064.755-4922.268) | 1.635(1.444-1.749) | 3883.420(3288.011-4222.780) | 1.049(0.888-1.141) | -1.851(-2.095-1.608) |
| Western Europe | 6968.778(6259.351-7388.562) | 1.813(1.628-1.922) | 8032.243(6636.608-8793.534) | 1.836(1.517-2.010) | -0.399(-0.752-0.045) |
| Southern Latin America | 1080.678(1011.000-1138.575) | 2.181(2.041-2.298) | 2872.408(2535.330-3121.091) | 4.243(3.745-4.611) | 3.054(2.807-3.301) |
| East Asia | 36347.387(31773.807-41338.631) | 2.986(2.610-3.396) | 16089.941(13321.990-19357.096) | 1.092(0.905-1.314) | -3.993(-4.626-3.355) |
| Central Asia | 3980.663(3632.442-4359.044) | 5.743(5.241-6.289) | 1686.777(1471.675-1925.608) | 1.761(1.536-2.010) | -4.076(-4.344-3.807) |
| Andean Latin America | 2658.604(2348.051-2966.227) | 6.998(6.180-7.807) | 2242.517(1847.181-2684.323) | 3.391(2.793-4.059) | -1.779(-2.154-1.403) |
| Tropical Latin America | 3997.397(3712.867-4336.268) | 2.620(2.434-2.842) | 6362.538(5569.140-6898.824) | 2.796(2.448-3.032) | 0.965(0.381-1.553) |
| Southern Sub-Saharan Africa | 2957.672(2629.460-3272.611) | 5.642(5.016-6.243) | 3847.691(3376.392-4325.692) | 4.791(4.205-5.387) | -0.012(-0.480-0.458) |
| Western Sub-Saharan Africa | 28256.480(22954.944-33552.658) | 14.629(11.884-17.371) | 25709.297(19871.848-32270.843) | 5.249(4.057-6.588) | -3.148(-3.324-2.972) |
| South Asia | 68845.247(58403.798-79147.722) | 6.296(5.341-7.239) | 43081.705(37809.429-48502.764) | 2.333(2.048-2.627) | -3.107(-3.201-3.013) |
| North Africa and Middle East | 14168.898(12243.757-17189.138) | 4.177(3.610-5.068) | 7251.617(6335.848-8196.005) | 1.164(1.017-1.316) | -3.778(-3.982-3.574) |
| Central Sub-Saharan Africa | 6966.605(5411.504-8513.339) | 12.675(9.846-15.489) | 5320.330(4173.062-6577.580) | 3.885(3.048-4.804) | -3.932(-4.113-3.751) |
| Central Latin America | 4472.180(4161.186-4808.879) | 2.720(2.531-2.925) | 4134.782(3634.339-4660.354) | 1.634(1.436-1.842) | -1.583(-2.181-0.982) |
| Eastern Sub-Saharan Africa | 25046.523(20743.794-30176.560) | 13.125(10.871-15.814) | 15217.104(12903.540-17658.054) | 3.571(3.028-4.144) | -4.344(-4.468-4.220) |
| Caribbean | 1339.772(1185.095-1534.845) | 3.796(3.358-4.349) | 1461.253(1278.657-1665.093) | 3.079(2.694-3.509) | -0.658(-0.872-0.443) |

GBD Global Burden of Disease Study, DALYs disability-adjusted life years, ASR Age-Standardized Rate

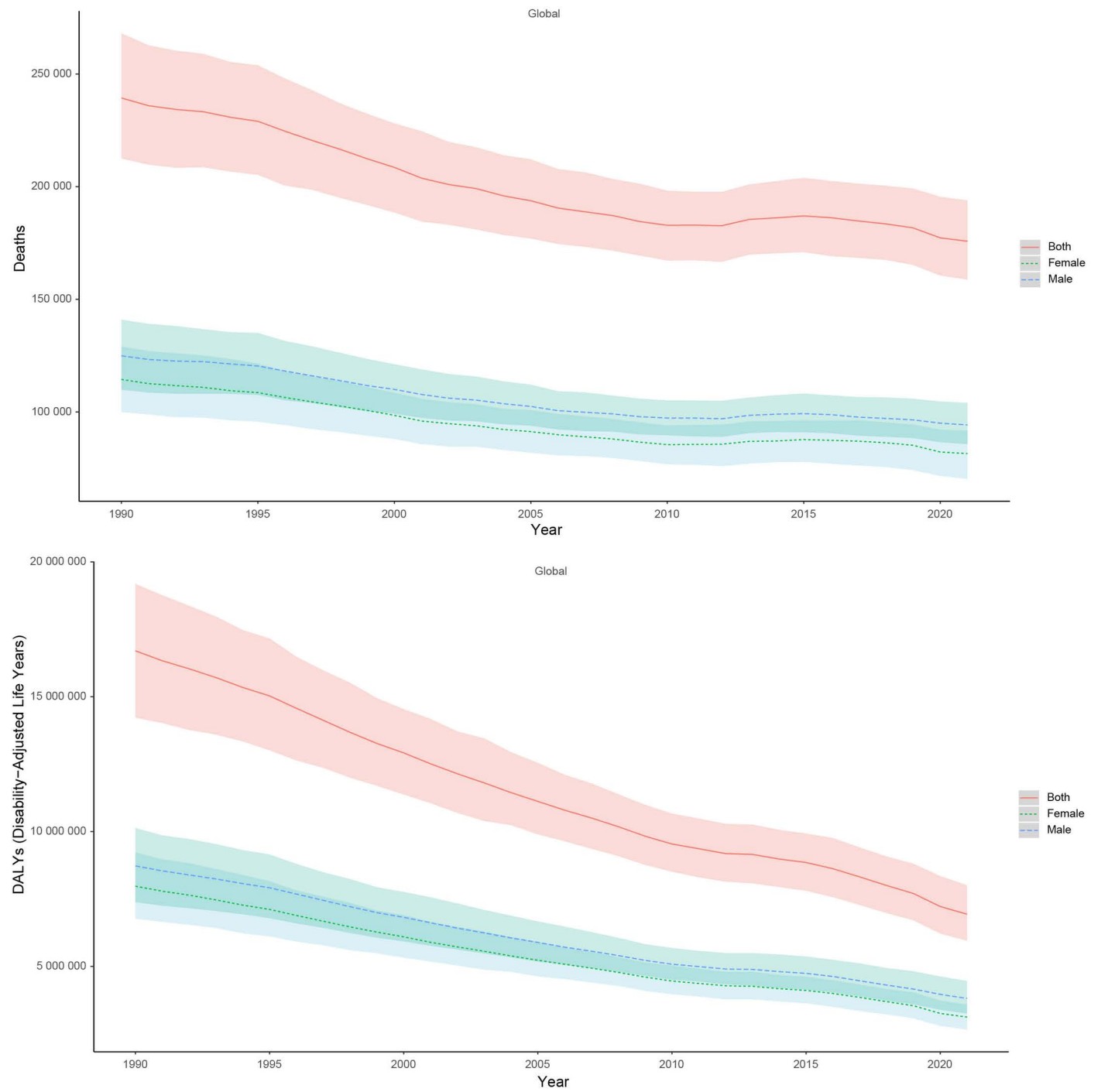

**Fig 1. The trends in numbers of DALYs and deaths for *Klebsiella pneumoniae* associated LRI burden from 1990 to 2021. DALYs, disability-adjusted life years.**

*pneumoniae* were observed in regions with lower SDI, whereas the lowest rates were seen in regions with higher SDI. From 1990 to 2021, age-standardized DALYs and death rates of LRI attributable to *Klebsiella pneumoniae* generally decreased as SDI increased. Figs 5b and 6b show age-standardized DALYs and death rates across all countries in 2021

a.

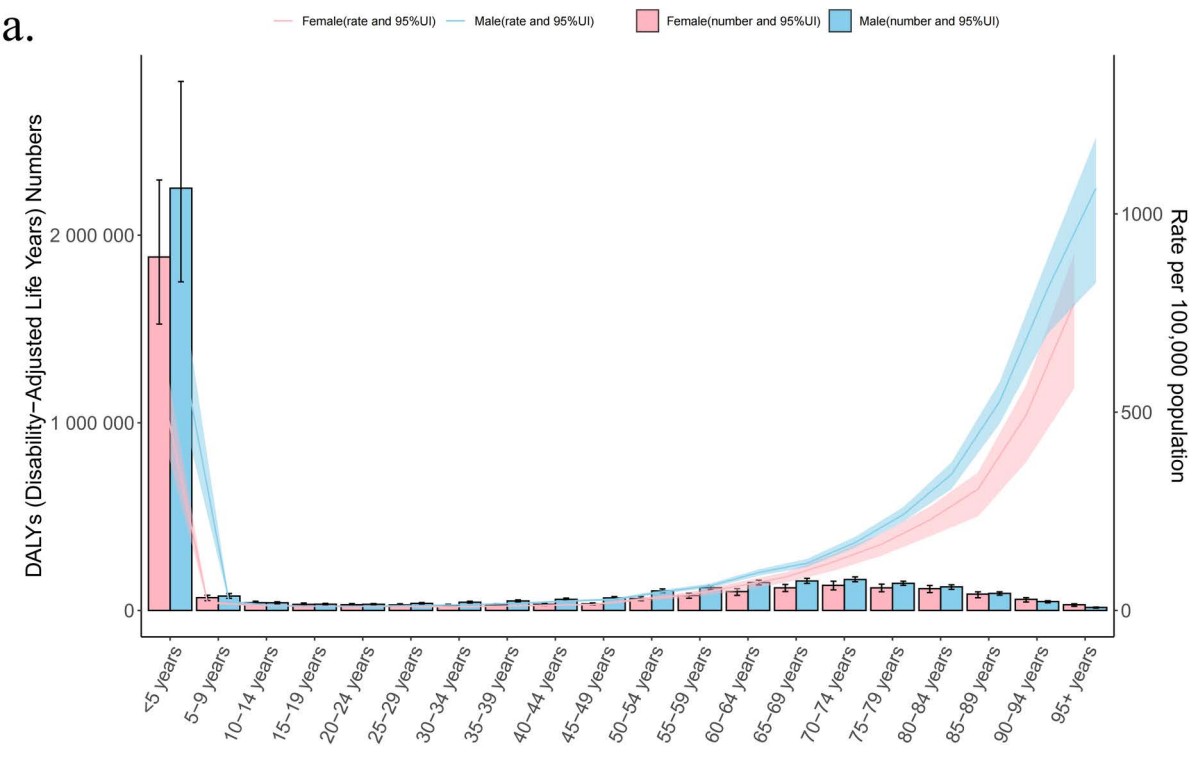

b.

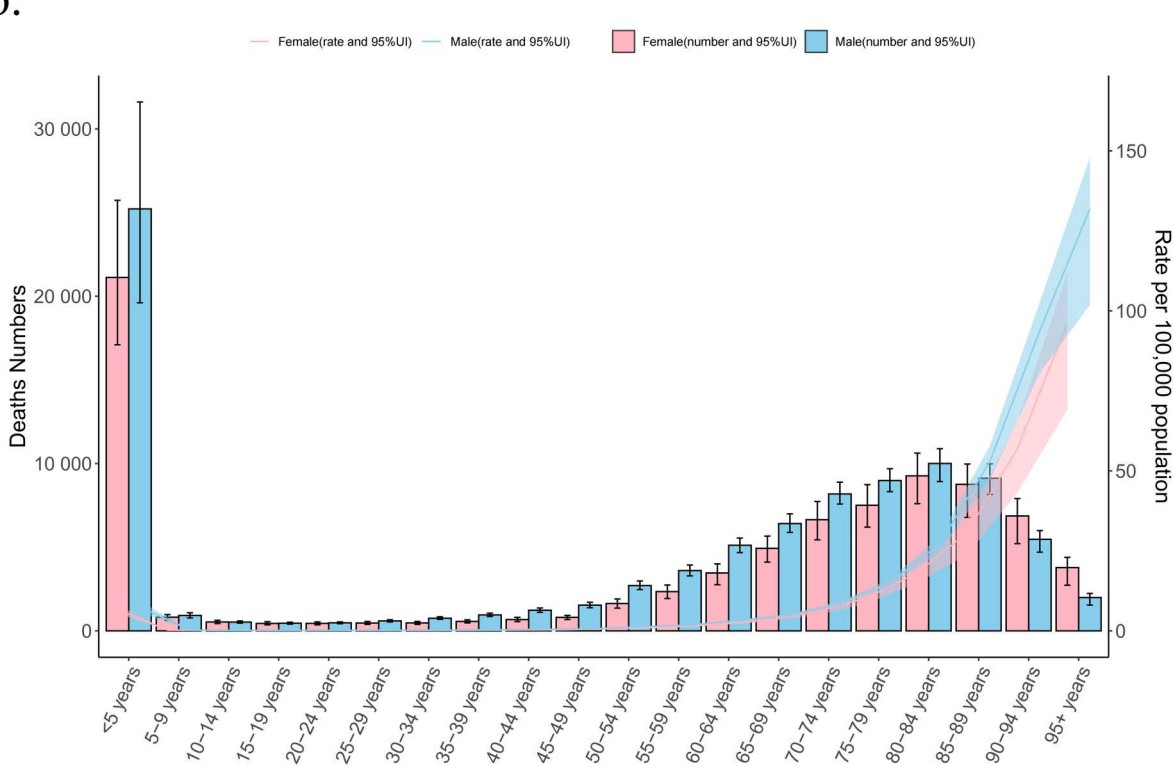

**Fig 2. Age-specific numbers and rates of DALYs (A) and deaths (B) for *Klebsiella pneumoniae* associated LRI burden by sex, 2021. DALYs disability-adjusted life years.**

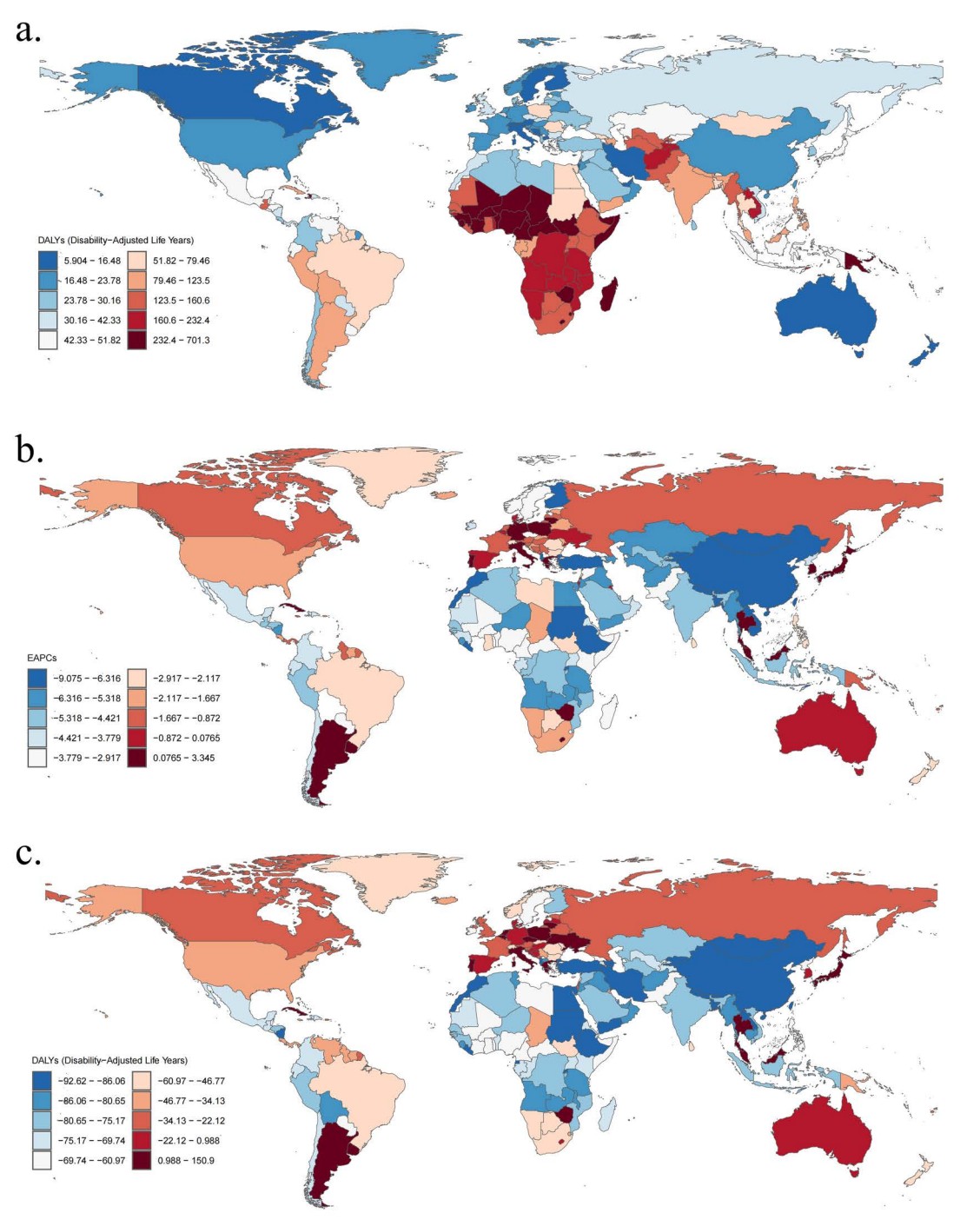

**Fig 3. The global disease burden of *Klebsiella pneumoniae* associated LRI for both sexes in 204 countries and territories.** (a) The number of DALYs in 2021; (b) The EAPC of DALYs from 1990 to 2019; (c) The change in DALYs cases between 1990 and 2019. DALYs disability-adjusted life years, EAPC estimated annual percentage change, LRI lower respiratory infections.

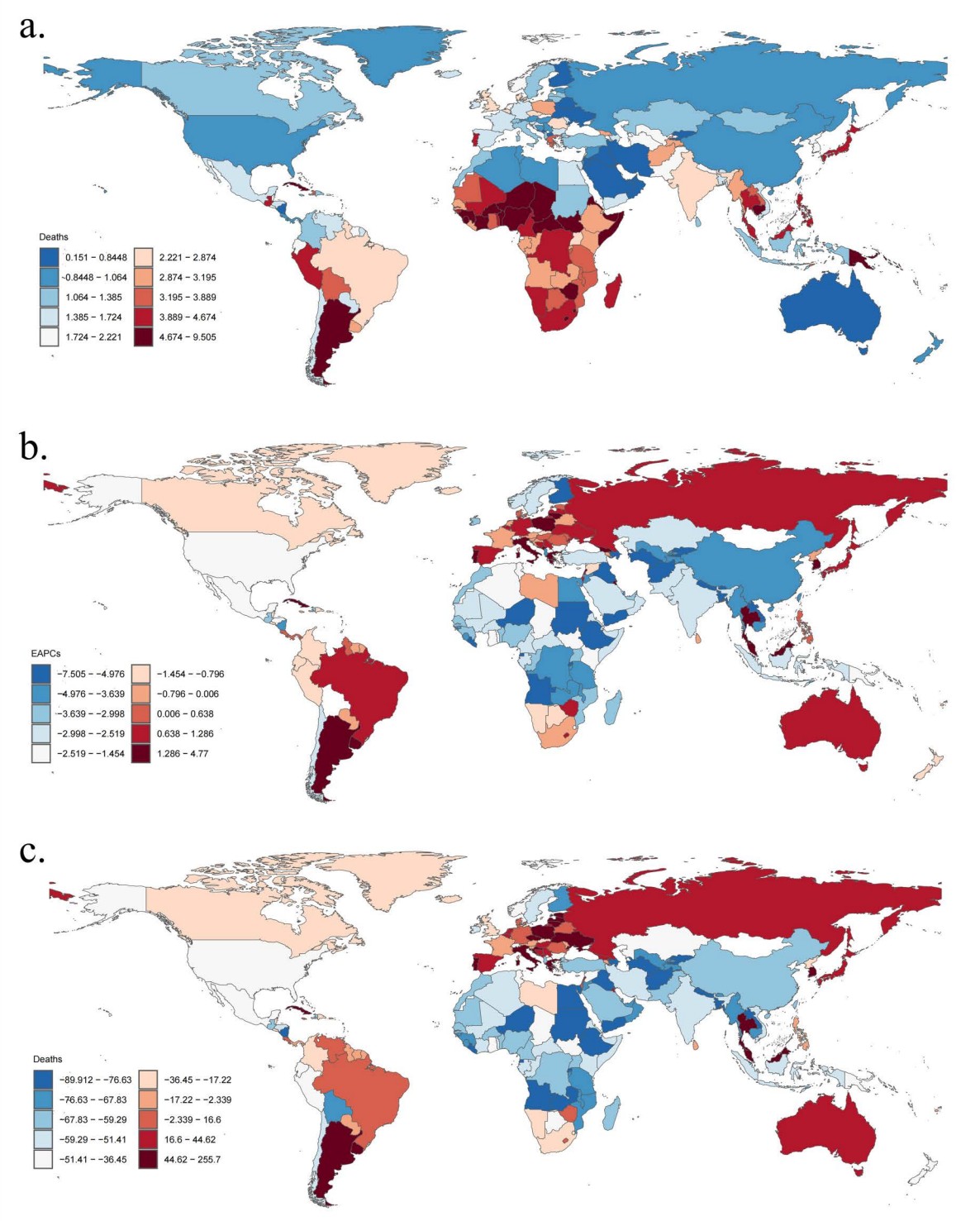

**Fig 4. The global disease burden of *Klebsiella pneumoniae* associated LRI for both sexes in 204 countries and territories. (a) The number of deaths in 2021; (b) The EAPC of deaths from 1990 to 2019; (c) The change in death cases between 1990 and 2019. EAPC estimated annual percentage change, LRI lower respiratory infections.**

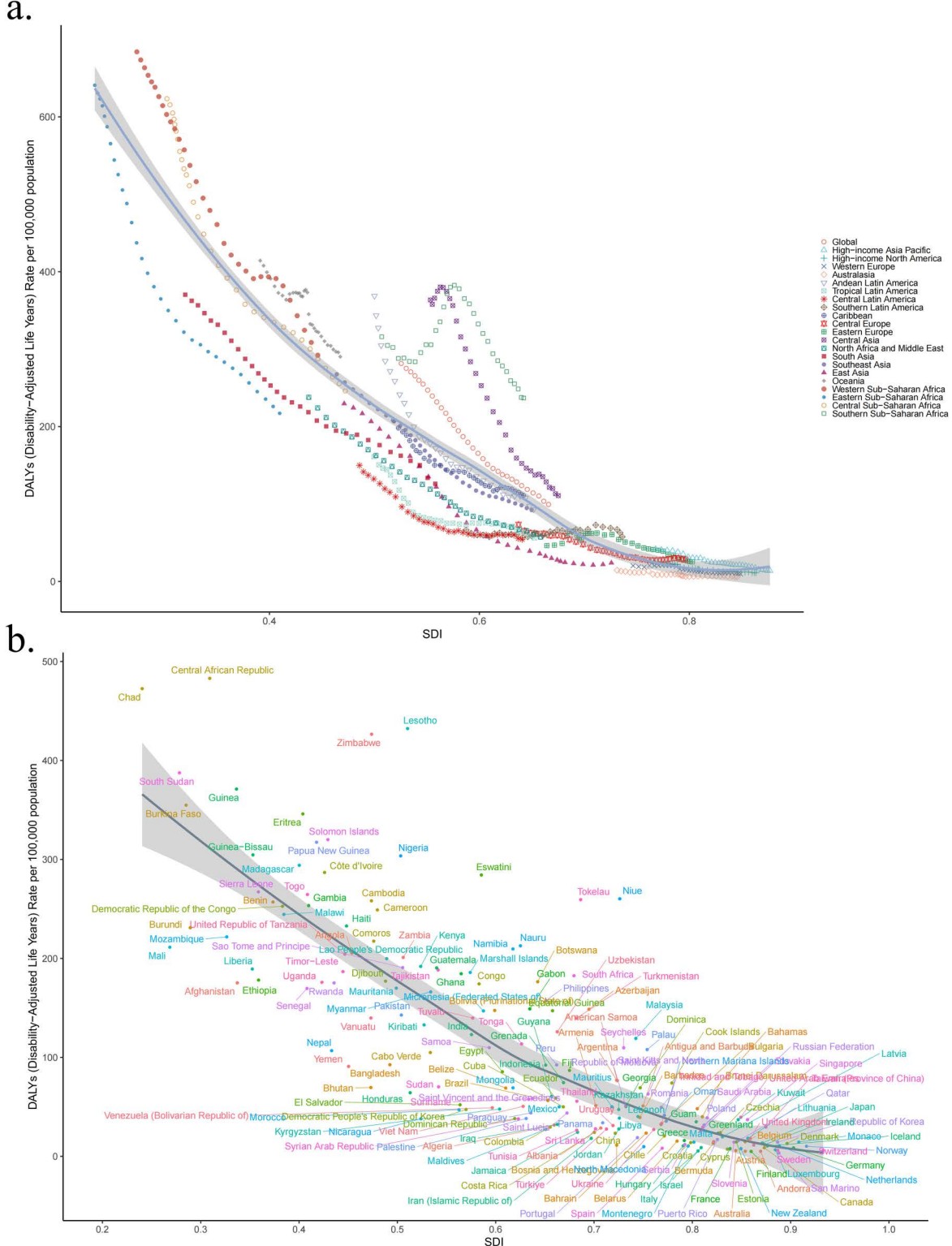

**Fig 5. DALYs rate and SDI across all regions between 1990 and 2021 (a). DALYs rate across all countries in 2021 by SDI (b). DALYs disability-adjusted life years, SDI socio-demographic index.**

a.

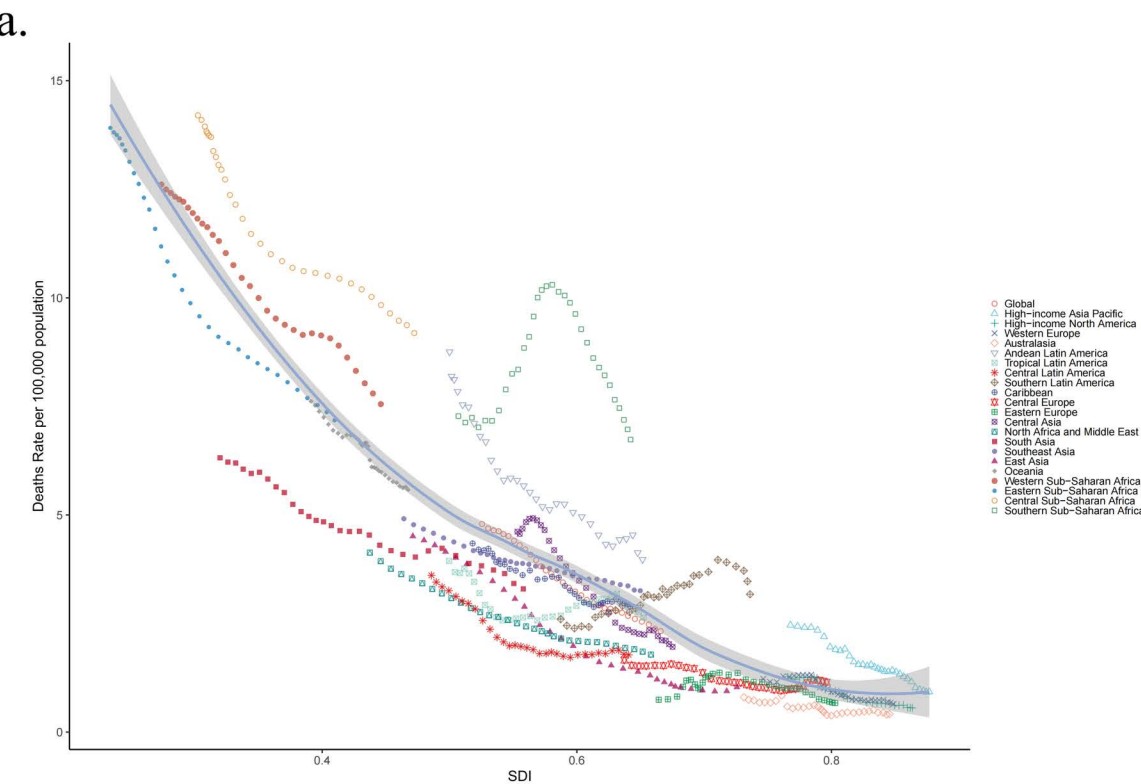

b.

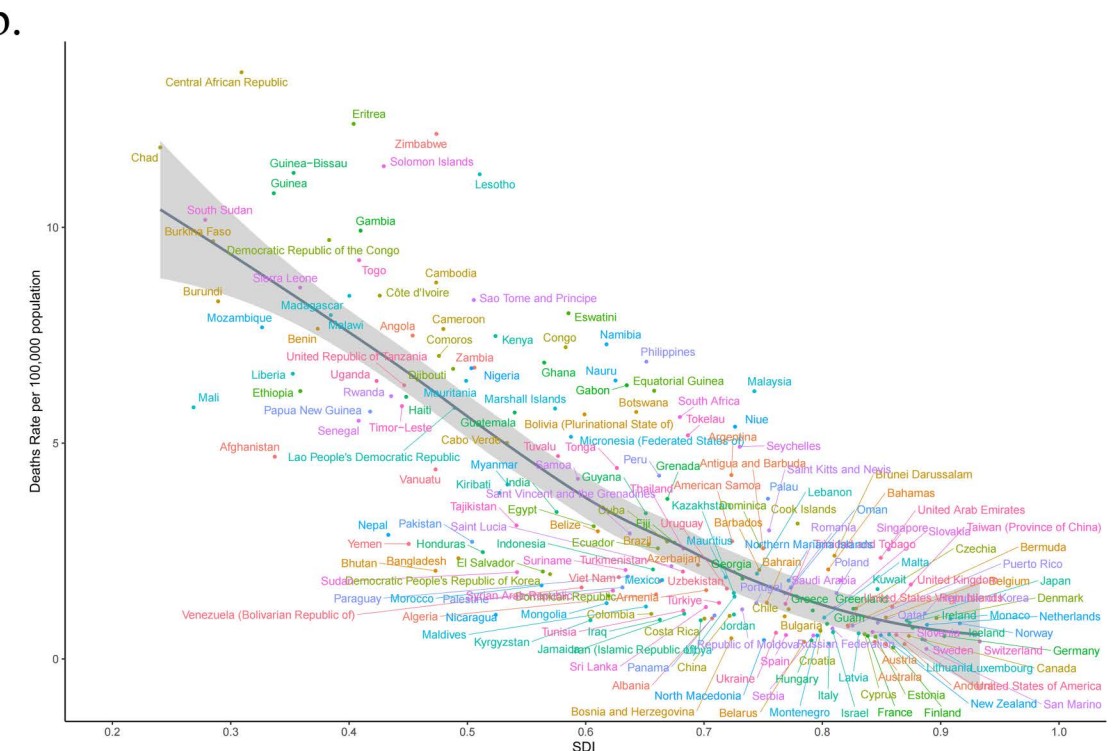

**Fig 6. Deaths rate and SDI across all regions between 1990 and 2021 (a). Deaths rate across all countries in 2021 by SDI (b). SDI socio-demographic index.**

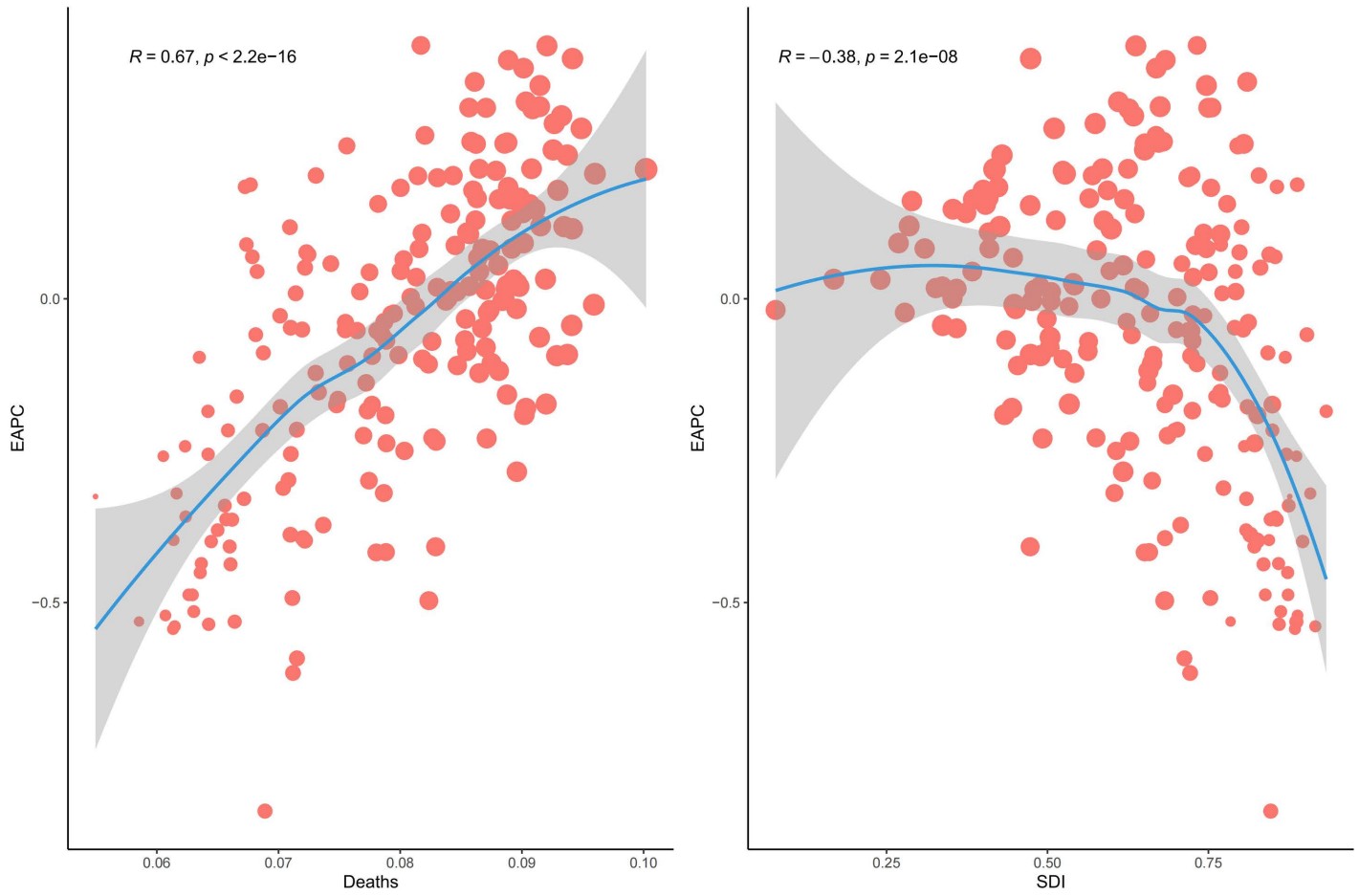

**Fig 7. Correlation between Estimated Annual Percentage Change (EAPC) of *Klebsiella pneumoniae* associated lower respiratory infection (LRI) and death rates, and Socio-demographic Index (SDI) in 2021.**

by SDI, generally decreasing with increasing SDI. Fig 7 shows a positive correlation between EAPC and death rates (R = 0.67, p < 2.2e-16), and a negative correlation between EAPC and SDI (R = −0.38, p = 2.1e-08).

## Discussion

This study provides a comprehensive overview of the global, regional, and national burden of *Klebsiella pneumoniae*-associated lower respiratory infections (LRI) from 1990 to 2021, using data from the Global Burden of Disease Study 2021 (GBD 2021). The significant decrease in both the number of disability-adjusted life years (DALYs) and deaths due to *Klebsiella pneumoniae*-associated LRI underscores the progress made in managing this critical public health issue. However, the variations in burden across different regions and demographics highlight ongoing challenges and areas for targeted interventions.LRI remain a major global health burden, accounting for a significant portion of morbidity and mortality worldwide. In 2021, it was estimated that there were 344 million (95% UI 325–364) LRI incidents and 2.18 million (1.98–2.36) deaths globally [5]. These figures highlight the pervasive impact of LRI on global health systems and the need for continuous monitoring and intervention strategies to mitigate their impact.*Klebsiella pneumoniae* is identified as one of the leading bacterial pathogens responsible for LRI mortality, ranking after Streptococcus pneumoniae and Staphylococcus aureus. In 2021, *Klebsiella pneumoniae*-associated LRI resulted in 175,783 deaths globally, reflecting its

significant contribution to the overall burden of LRI [5]. This pathogen's ability to cause severe infections, coupled with its high mortality rate, underscores the importance of addressing *Klebsiella pneumoniae* in global health strategies.Our study found higher ASR and numbers of DALYs and deaths in both males and females under 10 years old and over 65 years old. This indicates that the very young and the elderly are particularly vulnerable to severe outcomes from *Klebsiella pneumoniae*-associated LRI. A recent study shows that key risk factors for children include poor breastfeeding, malnutrition, and pollution, while for the elderly, smoking, alcohol consumption, and comorbidities are the main risks [16]. Targeted interventions and preventive measures should be prioritized for these age groups to reduce the disease burden.

Western Sub-Saharan Africa and Oceania, including countries like the Central African Republic, Niger, and Zimbabwe, had the highest burden of *Klebsiella pneumoniae*-associated LRI. Conversely, East Asia, Australasia, Eastern Europe, Western Europe, and High-income North America, including countries like Finland, Austria, and Germany, had the lowest burden. Regions with lower Socio-Demographic Index (SDI) values experienced higher disease burdens, highlighting the need for better healthcare infrastructure and access. High burden areas struggle with limited healthcare, poverty, and antimicrobial resistance, while low- and middle-income countries (LMICs) need improved healthcare systems and public health measures, supported by international collaboration [17].

The COVID-19 pandemic and the associated non-pharmaceutical interventions (NPIs) significantly impacted the transmission dynamics of respiratory infections, including *Klebsiella pneumoniae*-associated LRI. Measures such as stay-at-home orders, school and community closures, and facemask mandates effectively reduced the incidence of various respiratory infections, which indirectly affected the burden of bacterial LRI [18,19]. The implementation of these NPIs led to a notable decrease in respiratory virus transmission, which are often precursors to secondary bacterial infections such as those caused by *Klebsiella pneumoniae*. This reduction in viral infections likely contributed to the observed decrease in *Klebsiella pneumoniae*-associated LRI incidents and deaths during the pandemic period [20]. However, as NPIs are relaxed, continuous monitoring is essential to understand the long-term effects on LRI burden and to prevent potential resurgence.

One of the critical challenges in managing *Klebsiella pneumoniae* infections is the high prevalence of antimicrobial resistance (AMR) [21]. Similar to Staphylococcus aureus, *Klebsiella pneumoniae* has developed resistance to multiple antibiotics, complicating treatment regimens and leading to poorer clinical outcomes [22]. The emergence of carbapenem-resistant *Klebsiella pneumoniae* (CRKP) is particularly concerning, as carbapenems are often used as last-resort antibiotics for treating multidrug-resistant infections. The spread of CRKP has been documented in various regions, contributing to increased morbidity and mortality [23]. This highlights the urgent need for robust antibiotic stewardship programs and the development of new antimicrobial therapies to combat resistant strains.

Currently, there is no specific vaccine for *Klebsiella pneumoniae*, which poses a significant gap in the prevention of these infections [24]. Drawing lessons from the development and implementation of vaccines for other pathogens, such as Streptococcus pneumoniae and RSV, future efforts should focus on exploring vaccine development for *Klebsiella pneumoniae* [25]. Advances in immunology and biotechnology may provide pathways to develop effective vaccines that can prevent *Klebsiella pneumoniae* infections, particularly in vulnerable populations such as infants, the elderly, and immunocompromised individuals [26]. In addition to vaccine development, novel therapeutic approaches are being explored to address *Klebsiella pneumoniae* infections. Improved diagnostic technologies, such as Next-Generation Sequencing (NGS), can aid in the early detection and appropriate treatment of infections, thereby reducing the reliance on broad-spectrum antibiotics and minimizing the development of resistance [27]. Furthermore, research into new antimicrobial agents and alternative therapies, such as bacteriophage therapy and immunomodulatory treatments, is crucial for providing effective options against resistant strains of *Klebsiella pneumoniae* [28,29].

Despite the progress made in reducing the global burden of *Klebsiella pneumoniae*-associated LRI, there is a need for continued research to understand the pathogen's transmission dynamics, mechanisms of resistance development, and effective public health interventions. Future studies should focus on elucidating the factors contributing to the persistence

and spread of *Klebsiella pneumoniae* in different settings, particularly in healthcare facilities where nosocomial infections are prevalent [30]. Moreover, research into the development of new therapeutic agents is crucial for reducing the burden of *Klebsiella pneumoniae* infections. In recent years, significant efforts have been made by pharmaceutical companies and small- to medium-sized enterprises to develop new antibiotics to combat *Klebsiella pneumoniae* and other multidrug-resistant Gram-negative bacteria (MDR-GNB). These efforts are supported by global initiatives such as CARB-X and the Infectious Diseases Society of America's (IDSA) '10 × 20' initiative, which aimed to have 10 new antibacterial agents approved by the FDA by 2020 [31,32]. β-lactam antibiotics include innovative cephalosporins and β-lactamase inhibitors, such as ceftolozane and the ceftazidime-avibactam combination, which show significant activity against Enterobacterales producing KPC and OXA-48 enzymes [33]. Additionally, meropenem-vaborbactam, a combination of a carbapenem antibiotic and a β-lactamase inhibitor, has shown efficacy in treating complicated urinary tract infections (cUTI) [34]. Cefiderocol, a cephalosporin with a siderophore-like property, inhibits bacterial cell wall synthesis and demonstrates enhanced stability against various β-lactamases, including AmpC, ESBLs, and MBL [35]. Plazomicin, a newly FDA-approved aminoglycoside antibiotic, disrupts bacterial protein synthesis by targeting the 30S ribosomal subunit and is effective against bacteria producing ESBLs and certain carbapenemases like KPCs and OXA-48 [36]. These new antibiotics provide crucial treatment options for infections that previously had limited alternatives, but it is essential to use them judiciously to prevent the acceleration of antibiotic resistance, which could compromise their future efficacy [37].

In conclusion, this study highlights the significant global reduction in *Klebsiella pneumoniae*-associated LRI burden from 1990 to 2021. However, the persistent and, in some cases, increasing burden in certain regions underscores the need for continued public health efforts and international collaboration to address the challenges posed by this pathogen. Enhanced surveillance, improved healthcare access, and innovative medical interventions are essential to sustain and further these gains, ultimately reducing the global impact of *Klebsiella pneumoniae*-associated LRI. Future efforts should prioritize the development of targeted vaccines, the implementation of effective antimicrobial stewardship programs, and the exploration of novel therapeutic options to combat resistant strains. By addressing these critical areas, we can improve health outcomes, reduce the burden of disease, and achieve more equitable healthcare for populations worldwide.

## Supporting information

**S1 Table. Global and national estimates of disability-adjusted life years (DALYs) due to *Klebsiella pneumoniae*-associated lower respiratory infections in 1990 and 2021.** The table includes DALY counts, age-standardized DALY rates (per 100,000 population), and estimated annual percentage changes (EAPCs) for 204 countries and territories. Additionally, global-level estimates are presented separately for males and females.
(XLSX)

**S2 Table. Global and national estimates of deaths due to *Klebsiella pneumoniae*-associated lower respiratory infections in 1990 and 2021.** The table includes death counts, age-standardized mortality rates (per 100,000 population), and estimated annual percentage changes (EAPCs) for 204 countries and territories. Additionally, global-level estimates are presented separately for males and females.
(XLSX)

## Acknowledgments

We thank the study participants and other members of the team.

## Author contributions

**Conceptualization:** Ping Xu.

**Data curation:** Juanjuan Li, A Fang Zuo, Ping Xu, Kaizhi Xu.

**Formal analysis:** Juanjuan Li, A Fang Zuo.

**Investigation:** Kaizhi Xu.

**Resources:** Liang Xu.

**Software:** Liang Xu.

**Supervision:** A Fang Zuo.

**Writing – original draft:** Kaizhi Xu.

**Writing – review & editing:** Kaizhi Xu.

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
