## [Decision Letter · Decision Letter 0]

10 Mar 2025

PONE-D-24-54026The global burden of Klebsiella pneumoniae associated lower respiratory infection in 204 countries and territories, 1990–2021: fndings from the Global Burden of Disease Study 2021PLOS ONE

Dear Dr. Xu,

Thank you for submitting your manuscript to PLOS ONE. After careful consideration, we feel that it has merit but does not fully meet PLOS ONE’s publication criteria as it currently stands. Therefore, we invite you to submit a revised version of the manuscript that addresses the points raised during the review process.

**The reviewers recommended that you make moderate amendments to your manuscript.<!--EndFragment Principally in the bacterial scientific  names. **

We look forward to receiving your revised manuscript.

Kind regards,

Victoria Pando-Robles, Ph.D.

Academic Editor

PLOS ONE

**Journal Requirements:**

1. When submitting your revision, we need you to address these additional requirements. Please ensure that your manuscript meets PLOS ONE's style requirements, including those for file naming. The PLOS ONE style templates can be found at https://journals.plos.org/plosone/s/file?id=wjVg/PLOSOne_formatting_sample_main_body.pdf and https://journals.plos.org/plosone/s/file?id=ba62/PLOSOne_formatting_sample_title_authors_affiliations.pdf 2. Thank you for stating the following in the Acknowledgments Section of your manuscript: We thank the study participants and other members of the team. We are grateful to our hospital for supporting this study. We note that you have provided funding information that is not currently declared in your Funding Statement. However, funding information should not appear in the Acknowledgments section or other areas of your manuscript. We will only publish funding information present in the Funding Statement section of the online submission form. Please remove any funding-related text from the manuscript and let us know how you would like to update your Funding Statement. Currently, your Funding Statement reads as follows:  The author(s) received no specific funding for this work. Please include your amended statements within your cover letter; we will change the online submission form on your behalf. 3. Thank you for uploading your study's underlying data set. Unfortunately, the repository you have noted in your Data Availability statement does not qualify as an acceptable data repository according to PLOS's standards. At this time, please upload the minimal data set necessary to replicate your study's findings to a stable, public repository (such as figshare or Dryad) and provide us with the relevant URLs, DOIs, or accession numbers that may be used to access these data. For a list of recommended repositories and additional information on PLOS standards for data deposition, please see https://journals.plos.org/plosone/s/recommended-repositories. 4. PLOS requires an ORCID iD for the corresponding author in Editorial Manager on papers submitted after December 6th, 2016. Please ensure that you have an ORCID iD and that it is validated in Editorial Manager. To do this, go to ‘Update my Information’ (in the upper left-hand corner of the main menu), and click on the Fetch/Validate link next to the ORCID field. This will take you to the ORCID site and allow you to create a new iD or authenticate a pre-existing iD in Editorial Manager. 5. We note that Figures 3 and 4 in your submission contain map images which may be copyrighted. All PLOS content is published under the Creative Commons Attribution License (CC BY 4.0), which means that the manuscript, images, and Supporting Information files will be freely available online, and any third party is permitted to access, download, copy, distribute, and use these materials in any way, even commercially, with proper attribution. For these reasons, we cannot publish previously copyrighted maps or satellite images created using proprietary data, such as Google software (Google Maps, Street View, and Earth). For more information, see our copyright guidelines: http://journals.plos.org/plosone/s/licenses-and-copyright. We require you to either present written permission from the copyright holder to publish these figures specifically under the CC BY 4.0 license, or remove the figures from your submission: a. You may seek permission from the original copyright holder of Figures 3 and 4 to publish the content specifically under the CC BY 4.0 license.   We recommend that you contact the original copyright holder with the Content Permission Form (http://journals.plos.org/plosone/s/file?id=7c09/content-permission-form.pdf) and the following text:“I request permission for the open-access journal PLOS ONE to publish XXX under the Creative Commons Attribution License (CCAL) CC BY 4.0 (http://creativecommons.org/licenses/by/4.0/). Please be aware that this license allows unrestricted use and distribution, even commercially, by third parties. Please reply and provide explicit written permission to publish XXX under a CC BY license and complete the attached form.” Please upload the completed Content Permission Form or other proof of granted permissions as an "Other" file with your submission. In the figure caption of the copyrighted figure, please include the following text: “Reprinted from [ref] under a CC BY license, with permission from [name of publisher], original copyright [original copyright year].” b. If you are unable to obtain permission from the original copyright holder to publish these figures under the CC BY 4.0 license or if the copyright holder’s requirements are incompatible with the CC BY 4.0 license, please either i) remove the figure or ii) supply a replacement figure that complies with the CC BY 4.0 license. Please check copyright information on all replacement figures and update the figure caption with source information. If applicable, please specify in the figure caption text when a figure is similar but not identical to the original image and is therefore for illustrative purposes only.The following resources for replacing copyrighted map figures may be helpful: USGS National Map Viewer (public domain): http://viewer.nationalmap.gov/viewer/The Gateway to Astronaut Photography of Earth (public domain): http://eol.jsc.nasa.gov/sseop/clickmap/Maps at the CIA (public domain): https://www.cia.gov/library/publications/the-world-factbook/index.html and https://www.cia.gov/library/publications/cia-maps-publications/index.htmlNASA Earth Observatory (public domain): http://earthobservatory.nasa.gov/Landsat:
http://landsat.visibleearth.nasa.gov/USGS EROS (Earth Resources Observatory and Science (EROS) Center) (public domain): http://eros.usgs.gov/#Natural Earth (public domain): http://www.naturalearthdata.com/

Reviewers' comments:

Reviewer's Responses to Questions

**Comments to the Author**

1. Is the manuscript technically sound, and do the data support the conclusions?

Reviewer #1: Yes

Reviewer #2: Partly

2. Has the statistical analysis been performed appropriately and rigorously? 

Reviewer #1: Yes

Reviewer #2: Yes

3. Have the authors made all data underlying the findings in their manuscript fully available?

Reviewer #1: Yes

Reviewer #2: Yes

4. Is the manuscript presented in an intelligible fashion and written in standard English?

Reviewer #1: Yes

Reviewer #2: No

5. Review Comments to the Author

**Reviewer #1:**  The authors summarize the global epidemiological burden of lower respiratory infections (LRI) attributable to Klebsiella pneumoniae from 1990 to 2021, using data from the Global Burden of Disease Study. This reviewer considers this study analyze data that completely support the conclusions, with an appropiate statistical analysis. This analysis encompasses the incidence of Klebsiella infections in the respiratory tract in a broad and complete manner.

**Reviewer #2:**  This study investigates the global epidemiological burden of lower respiratory infections attributable to Klebsiella pneumoniae from 1990 to 2021, using data from the Global Burden of Disease Study (GBD) 2021.

The information presented in the manuscript is relevant, as it highlights the reduction in Klebsiella pneumoniae-associated LRI burden as well as the mortality caused by these bacteria from 1990 to 2021 around the world.

However, the way it is presented is too confusing and the wording of the manuscript is difficult to follow. Specific comments cannot be made because there is no numbering on each line of the manuscript, different font and paragraph styles are observed. Scientific names are misspelled (without italics), there are spaces between sentences and the tables have abbreviations that have no meaning in the text (for example, ASR).

In addition, the manuscript describes two objectives:

“This study aims to evaluate the trends and disparities at global, regional, and national levels, with a particular focus on Klebsiella pneumoniae” and “this study aims to provide a comprehensive analysis of the epidemiology of KP-associated LRIs from 1990 to 2021.”

I consider that the second objective is the one that defines the work carried out since the first one mentions that it evaluates the trends and disparities at global, regional, and national levels, with a particular focus on Klebsiella pneumoniae, when in reality it only focuses on K. pneumonia and not on other bacterial species.

Moreover, mentions the same results in different paragraphs,

“…Conversely, the regions with the lowest age-standardized DALYs and death rates of LRI attributable to Klebsiella pneumoniae were Australasia, High-income North America, Eastern Europe and East Asia…”

“… the most pronounced decreases in DALYs and death cases of LRI attributable to Klebsiella pneumoniae were observed in East Asia, whereas Southeast Asia, Southern Latin America, and Central Europe showed the most significant increases…”.

6. PLOS authors have the option to publish the peer review history of their article (what does this mean? ). If published, this will include your full peer review and any attached files.

**Do you want your identity to be public for this peer review?** For information about this choice, including consent withdrawal, please see our Privacy Policy .

Reviewer #1: No

Reviewer #2: No

---

## [Author Response · Author response to Decision Letter 1]

20 Mar 2025

Responses to the Editor's Comments:

1. Compliance with PLOS ONE Formatting Requirements

Comment:

Please ensure that the manuscript meets PLOS ONE's style requirements.

Response:

We have reviewed and revised the manuscript to fully comply with PLOS ONE' s formatting guidelines. All headings, text formatting, and file naming conventions have been adjusted according to the provided templates. We have also moved the results content earlier in the manuscript, directly before the figures and tables, in accordance with PLOS ONE's guideline that each figure caption should appear immediately after the paragraph in which it is first cited, and tables should be included directly after their first citation.Additionally, we confirm that the author order is correct, with Juanjuan Li as the first author and Kaizhi Xu as the corresponding author. The corresponding author' s email address has been listed accordingly in the manuscript.

2. Funding Statement Correction

Comment:

Funding-related information should not be included in the Acknowledgments section.

Response:

We have removed funding-related text from the Acknowledgments section. Our Funding Statement remains unchanged:

3. Data Availability Update

Comment:

The data repository previously used does not meet PLOS ONE' s standards.

Response:

To comply with PLOS ONE' s data availability requirements, we have uploaded the minimal dataset necessary to replicate our study' s findings to Figshare. The dataset is publicly accessible at the following DOI: https://doi.org/10.6084/m9.figshare.28616111, We have updated the Data Availability Statement in the manuscript accordingly.

4. ORCID iD Validation for Corresponding Author

Comment:

The corresponding author must have a validated ORCID iD in Editorial Manager.

Response:

The corresponding author's ORCID iD (0009-0005-2447-4491) has been successfully validated in Editorial Manager.

5. Copyright Compliance for Figures 3 and 4 (Map Images)

Comment:

Figures3 and 4 may contain copyrighted map images that must comply with the CC BY 4.0 license.

Response:

We confirm that the maps in Figures 3 and 4 were generated using the rnaturalearth R package, which provides access to Natural Earth map data. Natural Earth data is in the public domain and is not subject to copyright restrictions.

Data Source: Natural Earth is an open-source geographic dataset designed for unrestricted use, and further details can be found at https://github.com/ropensci/rnaturalearth.

Copyright Status: Natural Earth data is widely used for mapping and does not require explicit copyright permission.

Acknowledgment: If required, we are willing to include a statement in the figure legends acknowledging Natural Earth as the data source.

6. Reference List Verification and Formatting

Comment:

Ensure that the reference list is complete, correctly formatted, and free of retracted papers.

Response:

We have carefully reviewed the reference list to ensure accuracy and adherence to Vancouver style formatting. No retracted articles were found. Minor formatting inconsistencies have been corrected.

Responses to the Reviewer' s Comments:

Reviewer #1 Comments

Comment: The authors summarize the global epidemiological burden of lower respiratory infections (LRI) attributable to Klebsiella pneumoniae from 1990 to 2021, using data from the Global Burden of Disease Study. This reviewer considers this study analyze data that completely support the conclusions, with an appropriate statistical analysis. This analysis encompasses the incidence of Klebsiella infections in the respiratory tract in a broad and complete manner.

Response:

We sincerely thank the reviewer for the positive assessment of our study and for recognizing the robustness of our data analysis. We have carefully reviewed the manuscript to ensure clarity and coherence and have also implemented the necessary revisions based on Reviewer #2's comments.

Reviewer #2 Comments

1. Clarity and Readability of the Manuscript

Comment: The way the manuscript is presented is too confusing, and the wording is difficult to follow. Specific comments cannot be made because there is no numbering on each line of the manuscript, different font and paragraph styles are observed.

Response:

We appreciate the reviewer' s feedback regarding clarity and formatting. To improve readability, we have made the following modifications:

Ensured consistency in font and paragraph styles throughout the manuscript, adhering strictly to PLOS ONE formatting guidelines.

Revised complex and unclear sentences to improve readability.

Added line numbers to facilitate easier reference for future revisions.

These changes enhance the clarity and presentation of the manuscript.

2. Formatting Issues: Scientific Names and Abbreviations

Comment: Scientific names are misspelled (without italics), and there are inconsistencies in spacing. Tables contain abbreviations (e.g., ASR) that are not defined in the text.

Response:

We have carefully reviewed and corrected all instances of scientific names to ensure proper formatting:

Klebsiella pneumoniae is now consistently italicized throughout the manuscript.

All abbreviations (e.g., ASR, DALYs, SDI, EAPC) are now properly defined at their first mention in both the main text and table captions.

Inconsistent spacing issues have been addressed to align with the journal's formatting standards.

These corrections ensure accuracy and improve the manuscript' s professional presentation.

3. Refining Study Objectives

Comment:

The manuscript presents two objectives:

“This study aims to evaluate the trends and disparities at global, regional, and national levels, with a particular focus on Klebsiella pneumoniae.”

“This study aims to provide a comprehensive analysis of the epidemiology of Klebsiella pneumoniae-associated LRIs from 1990 to 2021.”

The second objective better defines the study, as the first one suggests an evaluation of trends in multiple bacterial species, while in reality, the study focuses only on Klebsiella pneumoniae.

Response:

We agree with the reviewer' s observation and have revised the study objective to ensure clarity:

The objective has been reworded to explicitly state that the study focuses on Klebsiella pneumoniae-associated LRIs rather than implying an evaluation of multiple pathogens.The relevant revision has been made in lines 51-55 of the introduction section.

4. Redundant Reporting of Results

Comment:

The manuscript presents overlapping results in different paragraphs, such as:

“…Conversely, the regions with the lowest age-standardized DALYs and death rates of LRI attributable to Klebsiella pneumoniae were Australasia, High-income North America, Eastern Europe, and East Asia…”

“… the most pronounced decreases in DALYs and death cases of LRI attributable to Klebsiella pneumoniae were observed in East Asia, whereas Southeast Asia, Southern Latin America, and Central Europe showed the most significant increases…”

Response:

We appreciate this observation and have carefully reviewed the Results section to eliminate redundancy. Specifically, we have:

Merged repetitive statements to present the findings more concisely.

Ensured that each paragraph presents unique information without unnecessary repetition.

These refinements improve the logical flow of the results, making the discussion more structured and easier to follow. The relevant modifications have been made in lines 231-233 of the Results section.

Final Remarks

We appreciate the reviewers' insightful feedback, which has significantly improved the clarity and quality of our manuscript. We have addressed all concerns and made the necessary revisions to enhance readability, formatting, and consistency.

---

## [Decision Letter · Decision Letter 1]

22 Apr 2025

The global burden of Klebsiella pneumoniae associated lower respiratory infection in 204 countries and territories, 1990–2021: fndings from the Global Burden of Disease Study 2021

PONE-D-24-54026R1

Dear Dr. Kaizhi Zhi Xu,

We’re pleased to inform you that your manuscript has been judged scientifically suitable for publication and will be formally accepted for publication once it meets all outstanding technical requirements.

Kind regards,

Victoria Pando-Robles, Ph.D.

Academic Editor

PLOS ONE

Additional Editor Comments (optional):

Reviewers' comments:

Reviewer's Responses to Questions

**Comments to the Author**

1. If the authors have adequately addressed your comments raised in a previous round of review and you feel that this manuscript is now acceptable for publication, you may indicate that here to bypass the “Comments to the Author” section, enter your conflict of interest statement in the “Confidential to Editor” section, and submit your "Accept" recommendation.

Reviewer #2: All comments have been addressed

2. Is the manuscript technically sound, and do the data support the conclusions?

Reviewer #2: Yes

3. Has the statistical analysis been performed appropriately and rigorously? 

Reviewer #2: Yes

4. Have the authors made all data underlying the findings in their manuscript fully available?

Reviewer #2: Yes

5. Is the manuscript presented in an intelligible fashion and written in standard English?

Reviewer #2: Yes

6. Review Comments to the Author

Reviewer #2: The objective has been reworded to explicitly state that the study focuses on Klebsiella pneumoniae-associated LRIs rather than implying an evaluation of multiple pathogens. The relevant revision has been made in lines 51-55 of the introduction section

Reviewer #2 Answer:

The modified target is actually on lines 45-46.

“This study aims to evaluate the global, regional, and national burden of Klebsiella pneumoniae-associated LRI from 1990 to 2021””

Reviewer #2 Comments:

Line 46. Delete comma

The manuscript has been corrected. The data presented on the global epidemiological burden of lower respiratory infections (LRI) attributable to Klebsiella pneumoniae from 1990 to 2021 are sound and supported by the analyses performed.

I recommend the manuscript for publication.

7. PLOS authors have the option to publish the peer review history of their article (what does this mean? ). If published, this will include your full peer review and any attached files.

**Do you want your identity to be public for this peer review?** For information about this choice, including consent withdrawal, please see our Privacy Policy .

Reviewer #2: No

---

## [Editor Report · Acceptance letter]

PONE-D-24-54026R1

PLOS ONE

Dear Dr. Xu,

I'm pleased to inform you that your manuscript has been deemed suitable for publication in PLOS ONE. Congratulations! Your manuscript is now being handed over to our production team.

Kind regards,

on behalf of

Dr. PLOS Manuscript Reassignment

Staff Editor

PLOS ONE